# Histamine Is Responsible for the Neuropathic Itch Induced by the Pseudorabies Virus Variant in a Mouse Model

**DOI:** 10.3390/v14051067

**Published:** 2022-05-17

**Authors:** Bing Wang, Hongxia Wu, Hansong Qi, Hanglin Li, Li Pan, Lianfeng Li, Kehui Zhang, Mengqi Yuan, Yimin Wang, Hua-Ji Qiu, Yuan Sun

**Affiliations:** 1State Key Laboratory of Veterinary Biotechnology, Harbin Veterinary Research Institute, Chinese Academy of Agricultural Sciences, 678 Haping Road, Harbin 150069, China; wangbing970112@163.com (B.W.); whx450650@163.com (H.W.); qhs199509@foxmail.com (H.Q.); lipan616@163.com (L.P.); lilianfeng@caas.cn (L.L.); zkhzhangkehui@163.com (K.Z.); yuanmengqi2019@163.com (M.Y.); 2Henan Institute of Science and Technology, College of Animal Science and Veterinary Medicine, Xinxiang 453003, China; hanglinLi@hotmail.com

**Keywords:** pseudorabies virus, itch, mouse, histamine, dorsal root ganglion

## Abstract

Pseudorabies virus (PRV) is the causative agent of pseudorabies (PR). It can infect a wide range of mammals. PRV infection can cause severe acute neuropathy (the so-called “mad itch”) in nonnatural hosts. PRV can infect the peripheral nervous system (PNS), where it can establish a quiescent, latent infection. The dorsal root ganglion (DRG) contains the cell bodies of the spinal sensory neurons, which can transmit peripheral sensory signals, including itch and somatic pain. Little attention has been paid to the underlying mechanism of the itch caused by PRV in nonnatural hosts. In this study, a mouse model of the itch caused by PRV was elaborated. BALB/c mice were infected intramuscularly with 10^5^ TCID_50_ of PRV TJ. The frequency of the bite bouts and the durations of itch were recorded and quantified. The results showed that the PRV-infected mice developed spontaneous itch at 32 h postinfection (hpi). The frequency of the bite bouts and the durations of itch were increased over time. The mRNA expression levels of the receptors and the potential cation channels that are relevant to the itch-signal transmission in the DRG neurons were quantified. The mRNA expression levels of tachykinin 1 (TAC1), interleukin 2 (IL-2), IL-31, tryptases, tryptophan hydroxylase 1 (TPH1), and histidine decarboxylase (HDC) were also measured by high-throughput RNA sequencing and real-time reverse transcription PCR. The results showed that the mean mRNA level of the HDC in the DRG neurons isolated from the PRV-infected mice was approximately 25-fold higher than that of the controls at 56 hpi. An immunohistochemistry (IHC) was strongly positive for HDC in the DRG neurons of the PRV-infected mice, which led to the high expression of histamine at the injected sites. The itch of the infected mice was inhibited by chlorphenamine hydrogen maleate (an antagonist for the histamine H1 receptor) in a dose-dependent manner. The mRNA and protein levels of the HDC in the DRG neurons were proportional to the severity of the itch induced by different PRV strains. Taken together, the histamine synthesized by the HDC in the DRG neurons was responsible for the PRV-induced itch in the mice.

## 1. Introduction

Pseudorabies virus (PRV), which is closely related to varicella-zoster virus (VZV) and herpes simplex virus type 1 (HSV-1), is a member of the *Alphaherpesvirinae* subfamily within the *Herpesviridae* family [1]. Aujeszky’s disease (AD) is caused by PRV, which was described and demonstrated by Aladár Aujeszky in 1902 [2]. However, the disease was first reported as “mad itch”, and it showed clinical severe itch in cattle [3]. Currently, the distinct natural reservoir of PRV is swine; however, PRV can also infect a wide range of mammals, such as mice, rabbits, sheep, goats, and even humans [4,5]. More than 100 years have passed since the “mad itch” disease was first reported in nonnatural hosts. However, the PRV TJ strain, which is a variant PRV currently prevalent in China, caused unusual pruritus in pigs, which are the natural hosts of PRV [6]. The mechanism that underlies the PRV-induced itch in mouse models can provide a basis for the understanding of the PRV-induced pruritus in pigs.

Itch is an “unpleasant” sensation that leads to the scratch reflex, and it has many similarities to pain. There are four categories of itch: the pruriceptive itch, the neurogenic itch, the neuropathic itch, and the psychogenic itch [7]. The pruriceptive itch originates in the skin and is caused by inflammatory disorders. The inflammatory disorders activate the pruriceptive primary afferent, and the itch signals are transmitted from the skin into the sensory neurons in the dorsal root ganglion (DRG). The neurogenic itch results from central-nervous-system (CNS) activation, without the necessary activation of the sensory nerve fibers. The neuropathic itch can originate at any point along the afferent pathway because of the damage to the nervous system caused by viral disease and/or the traumatic nerve injury of the PNS or the CNS, such as peripheral neuropathies (e.g., postherpetic itch), multiple sclerosis, and nerve compression or irritation. The psychogenic itch is related to psychological or psychiatric disorders, such as itch-associated with delusions of parasitosis, stress, and depression.

Itch is mediated by peripheral somatosensory neurons that are termed “pruriceptors”, which sense and respond to pruritogens. Tachykinin 1 (TAC1), interleukin 2 (IL-2), IL-31, and tryptases [tryptase alpha/beta 1 (TPSAB1), tryptase beta 2 (TPSB2), and tryptase gamma 1 (TPSG1)] have been identified as pruritogens, which can evoke itch signals [8]. TAC1 is a kind of neuropeptide that elicits biting and scratching in mice [9]. Moreover, IL-2, IL-31, and tryptase (a kind of serine protease) can also elicit itch [10,11]. Histamine or 5-hydroxytryptamine (5-HT) is synthesized by histidine decarboxylase (HDC) and tryptophan hydroxylase 1 (TPH1), respectively, which can evoke scratching. Histamine is a well-established pruritogen, and it has been regarded as the main target for antipruritic therapies. Histamine produces itch in humans accompanied by skin reactions (wheal and flare) [12,13]. Itch can also be elicited in the mice that are injected intradermally with histamine [14,15].

The DRG neurons are composed of the cell bodies of genetically distinguishable primary afferent neurons, and they are crucial structures in sensory transduction and modulation, including in itch transmission. There is a wide range of molecules that are involved in the transmission of itch signals in the DRG neurons [7]. Histamine receptor H1 (HRH1) is expressed in itch-sensing DRG neurons and it mediates histaminergic itch. The family of Mas-related G-protein-coupled receptors (Mrgprs) mediates nonhistaminergic itch, and it includes MrgprA3, MrgprD, and MrgprX1 [7,12]. Additionally, transient receptor potential cation channel subfamily V member 1 (TRPV1), TRPA1, TRPV4, and TRPM8 are also involved in the transmission of itch signals in the DRG. Because of their important roles in signal transduction, the DRG neurons are widely used in itch research. 

Some bacterial and viral infections, including PRV, can cause itch. However, the underlying mechanisms that lead to the activation of pruriceptors are not well understood. In this study, a mouse model of PRV TJ-induced itch was established, and the severity of the itch in the mice infected with the PRV TJ, SC, or Bartha-K61 strains was also compared. We showed that the histamine synthesized by the HDC in the DRG neurons was responsible for the PRV-induced itch in the mice.

## 2. Materials and Methods

### 2.1. Viruses and Cells

The PRV Bartha-K61 strain (GenBank accession No. JF797217.1) is a widely used attenuated PRV vaccine [16,17]. The PRV SC strain (GenBank accession No. KT809429.1) is a classical virulent PRV strain [18,19]. The PRV TJ strain (GenBank accession No. KJ789182.1) was isolated from a Bartha-K61-vaccinated pig farm in Tianjin, China, in 2012, and it is more virulent than PRV SC. All the PRV strains are saved at the Harbin Veterinary Research Institute (HVRI), at the Chinese Academy of Agricultural Sciences (CAAS). PK-15 cells (ATCC, CCL-33) were cultured in Dulbecco’s modified Eagle’s medium (DMEM) (Gibco, Carlsbad, CA, USA) supplemented with 5% fetal bovine sera (FBS) (Hyclone, Logan, UT, USA), 100 mg/mL streptomycin, and 100 IU/mL penicillin at 37 °C and 5% CO_2_. All the PRV strains were propagated and titrated on PK-15 cells.

### 2.2. Virus Titration

PK-15 cells were seeded into 96-well plates, infected with 10-fold serially diluted PRV suspensions, and cultured at 37 °C, 5% CO_2_, for 72 h. The PRV-infected cells were detected by indirect immunofluorescence assay (IFA), and the number of fluorescent wells was counted, as described previously [19]. Then, the viral titers were calculated by using the Reed and Muench method, and they were expressed as the median tissue culture infectious dose (TCID_50_) [20].

### 2.3. Behavioral Observation of the PRV-Infected Mice

All the animal experiments were conducted under a protocol (210603-02) approved by HVRI.

Six-week-old specific pathogen-free (SPF) BALB/c mice were used in this study, and all the mice were housed in a pathogen-free environment at 22 to 25 °C and with an ad libitum water and food supply. To observe the behaviors of the PRV-infected mice, twelve mice were divided into two groups, and one group was intramuscularly (i.m.) injected with 100 μL of inoculum containing 10^6^ TCID_50_ of PRV TJ into the left hindlimb muscle, and the other group was mock inoculated (100 μL of medium only). To avoid behavior changes caused by different surroundings, mice were videotaped in an isolator from 0 h postinfection (hpi) to the moribund state of 56 hpi [21,22,23]. Additionally, high-resolution video and analysis techniques at slow playback speeds were used to discriminate between biting and licking. The severity of the itch in the mice was determined by the frequency of bite bouts and by the durations of itch. When the mice started itching, the bite bouts were counted for 30 min at 1.5 h intervals, and the durations of itch were also counted for 30 min at 3.5 h intervals [21,22,23]. The frequency of the bite bouts and the durations of itch were counted by a blindly trained observer on the basis of videos. 

### 2.4. Isolation of the DRG Neurons and Skin from the PRV-Infected Mice

A total of 33 6-week-old SPF BALB/c mice were i.m. injected with 100 μL of 10^6^ TCID_50_/_mL_ PRV TJ suspensions into the left hindlimb muscle, and the same number of mice was injected with 100 μL of DMEM as the mock. To isolate the DRG neurons located on the left side of the spinal cord, three mice from the PRV TJ group and the mock group were euthanized at 0, 2, 8, 14, 20, 26, 32, 38, 44, 50, and 56 hpi. After removing the fur, muscles, and the dorsal portion of the spine, the spinal cord was exposed. The spinal cord was removed with forceps, and the DRG neurons were collected [24]. Skin at the injection sites was also collected.

### 2.5. RNA-Seq Analysis

For the whole transcriptomic analysis, total RNAs were extracted from the DRG neurons isolated from the mock or PRV TJ-infected mice by using RNAiso Plus reagent. All the sample analyses were carried out in triplicates. Genome-wide differential gene expression was analyzed by using RNA deep sequencing by BGI Genomics (Shenzhen, China). Poly(A) plus RNAs were sequenced by using the Illumina HiSeq 2500 platform. The high-quality reads were further mapped to the reference genome of the mice by using Hisat2 (https://daehwankimlab.github.io/hisat2/, accessed on 8 June 2021), and the transcripts were assembled by using Stringtie (http://ccb.jhu.edu/software/stringtie/, accessed on 8 June 2021).

### 2.6. Reverse Transcription-Quantitative PCR (RT-qPCR)

After isolation from the spinal cords of the mice at 0, 2, 8, 14, 20, 26, 32, 38, 44, 50, and 56 hpi, the DRG neurons were ground in liquid nitrogen. Total RNAs from the DRG were extracted by using RNAiso Plus (catalog No. 9108Q; TaKaRa, Beijing, China), according to the manufacturer’s protocols. Total RNAs were also extracted from the skin in the same way. Then, the total RNAs were reverse transcribed into cDNA by using the HiScript II Reverse Transcriptase (catalog No. R201-02; Vazyme, Nanjing, China), according to the instructions. GADPH was amplified as a normalization control. RT-qPCR was performed by using the 2×ChamQ SYBR qPCR Master Mix (catalog No. Q311-03; Vazyme, Nanjing, China). Forty cycles of amplification were performed, which included sequential denaturation at 95 °C for 10 s, annealing at 60 °C for 30 s, and extension at 72 °C for 1 min. All the primers used in this experiment are listed in Table 1, and all the samples were analyzed in triplicates.

### 2.7. Effects of Chlorphenamine Hydrogen Maleate Treatment on the PRV TJ-Infected Mice

The chlorphenamine hydrogen maleate powder (catalog No. BP081; Sigma-Aldrich, Darmstadt, Germany) was dissolved in DMEM. Thirty 6-week-old SPF BALB/c mice were divided into five groups (*n* = 6): four groups were i.m. injected with 100 μL of inoculum containing 10^6^ TCID_50_ of PRV TJ into the left hindlimb muscle, and one group was injected with 100 μL of DMEM as a control. The mice of the four PRV TJ-infected groups were i.m. injected with 0, 10, 30, or 60 mg/kg chlorphenamine hydrogen maleate solution in the left hindlimb muscle at 44 and 50 hpi. Twenty minutes later, the itching behavior was recorded.

### 2.8. Hematoxylin and Eosin Staining and Immunohistochemistry (IHC)

At 56 hpi, all the mice were euthanized for pathological examinations. The left legs and the lumbosacral DRG neurons of the mice were freshly collected by using scissors and forceps, were fixed in 4% paraformaldehyde for 48 h, followed by 70% ethanol, and were embedded in paraffin, cut into 5-μm sections, and mounted onto glass slides. The sections were stained with hematoxylin and eosin stain [25]. For the immunohistochemical examination, the sections were blocked in 10% normal goat serum, incubated with a rabbit antihistamine antibody (catalog No. H7403-.2ML; Sigma-Aldrich, Darmstadt, Germany) overnight at 4 °C, washed three times in TBST buffer, incubated in SignalStain Boost IHC detection reagent (HRP, Rabbit) (catalog No. 8114P; Sinopharm, Beijing, China) for 30 min at room temperature, washed 3 times, and stained [26].

### 2.9. Comparison of Itch in the Mice Infected with Different PRV Strains

Twenty-four mice were divided into four groups (*n* = 6): three groups were i.m. injected with 10^5^ TCID_50_ of PRV TJ, SC, or Bartha-K61 into the left hindlimb muscle, respectively. The last group was injected with 100 μL of DMEM as a control. All the mice were videotaped, and the bite bouts and durations of itch were counted at 56 hpi [21,22,23]. Additionally, the DRG neurons were isolated, and the mRNAs of the HDC were quantified by using RT-qPCR.

### 2.10. qPCR

The mice infected with 10^5^ TCID_50_ of PRV TJ, SC, Bartha-K61, or DMEM were euthanized at 56 hpi. The DRG neurons were collected, and DNA was extracted by using the MagaBio plus virus DNA purification kit (catalog No. 9109; BioFlux, Beijing, China), according to the manufacturer’s protocols. The genomic copies of PRV were quantified on the basis of a previously described method [27].

### 2.11. Statistical Analysis

All the experiments were performed in triplicates, and the statistical significance was analyzed by Student’s *t* test in Prism v8.0 software (GraphPad, San Diego, CA, USA). Differences were considered significant if the unadjusted *p*-value was less than 0.05.

## 3. Results

### 3.1. The Severity of the Itch Caused by PRV TJ Infection Gradually Increased with Time

To quantify the severity of the itch, the mice that were injected with PRV TJ or DMEM were videotaped by using two high-resolution videos, respectively. The behaviors of the mice were recorded constantly from 0 to 56 hpi. As for the biting action, there were no intuitive differences between natural and itchy bites. However, in terms of the biting frequency, natural bites were occasional, while itchy bites were regular and frequent in the PRV TJ-infected mice. In addition, the natural bites were transient, but the durations of the itchy bites increased as the clinical signs progressed. On the basis of this criterion, the frequency of the bite bouts and the durations of itch were quantified. The results showed that the mice infected with PRV TJ did not exhibit spontaneous biting from 0 to 30 hpi; however, from 32 hpi, the mice started itching. The infected mice bit the injection sites about 10 times per 30 min at 32 hpi, spontaneously, and the frequency of the bite bouts increased gradually, reached about 62 times per 30 min at 56 hpi (Figure 1A), and was significantly different between the PRV TJ-infected group and the control group (*p* < 0.01). Additionally, the duration of the biting was counted during the 30 min observation period. We found that the duration of the itch remained at the same level as the mock group until 28 hpi. 

None of the PRV TJ-infected mice exhibited itch until 28 hpi, and the average duration of the itching was about 5 s per 30 min at 32 hpi. The duration rose gradually over time, and it peaked at about 90 s per 30 min at 52 hpi (Figure 1B). The mice injected with DMEM did not exhibit spontaneous biting during the whole experiment period.

The mice were also photographed at 0, 32, 44, and 56 hpi. The legs of the infected mice showed no signs of tissue damage from 0 to 32 hpi, although the mice infected with PRV TJ started biting at 32 hpi. At 44 hpi, because of the long-term biting, severe tissue damage appeared at the inoculated sites, and the muscle and skin were detached. At 56 hpi, the left legs of the mice were shriveled. For the control mice, no injuries were found in the legs during the whole experiment period (Figure 1C).

### 3.2. Different Expression Profiles of the Molecules Relevant to Itch-Signal Transmission Were Noted in the DRG Neurons

The molecules that are relevant to itch-signal transmission were detected by using RNA-seq and RT-qPCR. The DRG neurons from the infected or the control mice were harvested at 56 hpi. On the heat map, the transcription levels of HRH1, MrgprA3, TRPV4, and TRPA1 were comparable between the two groups at 56 hpi. However, the mRNA expression levels of MrgprX1, MrgprD, TrpA1, and TrpV1 were reduced to varying degrees (Figure 2A).

The mRNA expression levels of the receptors and channels in the DRG neurons were measured by using RT-qPCR. As shown in Figure 2B, the mRNA expression of MrgprX1 did not change significantly from 0 to 50 hpi; however, it decreased at 56 hpi, which was at approximately half of the mock group. The mRNA expression level of MrgprA3 also did not change significantly over the test period. Interestingly, the mRNA level of MrgprD increased gradually, from 0 to 32 hpi, it remained at a plateau from 32 to 38 hpi, its maximum expression in the PRV TJ-infected group was higher than that in the mock group, and it returned to basal level. Finally, the mRNA expression level of MrgprD was reduced by approximately 50%. In addition, the mRNAs of the receptors TRPA1, TRPM8, TRPV1, and TRPV4 were also measured by using RT-qPCR. Among them, TRPV1 and TRPV4 contribute to histamine-mediated neuronal activation. TRPA1 is extensively involved in the neuronal activation in nonhistaminergic itch. TRPM8 is another receptor that modulates or mediates itch. The results showed that the mRNA level of TRPA1 in the PRV TJ-infected group did not change from 0 to 32 hpi, increased transitorily from 32 to 38 hpi, then decreased gradually, and was lower than that of the mock group. The mRNA expression levels of the two TRP cation channels that mediate histaminergic itch were different. The TRPV1 mRNA in the infected group was consistent with that in the mock group from 0 to 20 hpi, then it increased gradually from 20 to 38 hpi, and it decreased from 38 to 56 hpi. The maximal TRPV1 mRNA level was about 2.7-fold higher than that of the control at 38 hpi. In contrast to the TRPA1 and TRPV1, the TRPV4 mRNA remained relatively stable, with changes only from 0 to 56 hpi. The TRPM8 mRNA was unchanged from 0 to 8 hpi, it then increased from 8 to 26 hpi, it remained at a plateau from 26 to 32 hpi (the highest mRNA level of TRPM8 was approximately 3-fold higher than that of the control), and it finally returned to a basal level.

### 3.3. The Expression of HDC Was Increased in the PRV TJ-Infected DRG Neurons

PRV can replicate and establish quiescent, latent infection in the PNS [28]. Moreover, PRV can induce a severe inflammation response in the DRG neurons. A previous study has shown that PRV in the neurons may trigger itch [29]. TAC1, IL-2, IL-31, TPSAB1, TPSB2, and TPSG1 are known as pruritogens [14,15]. HDC and TPH1 are the key enzymes of pruritogen synthesis. The mRNA levels of pruritogens and their correlated molecules were detected by using RNA-seq. The HDC transcription level of the PRV TJ-infected group was significantly higher than that of the control group (Figure 3A).

HDC mRNA was detected by using RT-qPCR in the DRG neurons of the PRV TJ or mock groups at 0, 2, 8, 14, 20, 26, 32, 38, 44, 50, and 56 hpi (Figure 3B). The mRNA level of the HDC in the PRV TJ group increased 1.7-fold compared to that in the control group at 20 hpi, then it returned to a normal level at 26 hpi, and, notably, it increased rapidly from 32 to 56 hpi. The maximum expression level in the infected group was approximately 25-fold higher than that in the mock group in the moribund state. The gB protein of the PRV TJ was detected in the DRG neurons by using IHC at 56 hpi (red arrows), and the protein expression level of the HDC (blue arrows) was also significantly higher than that of the control (Figure 3C).

### 3.4. Histamine Produced in the DRG Neurons Contributed to PRV-Induced Itch and Could Be Inhibited by Chlorpheniramine

HDC catalyzes the conversion of histidine to histamine, which is a “gold standard” itch mediator [7]. A significantly higher expression level of histamine was detected in the PRV TJ-infected group compared to the control group (Figure 4A). The main source of histamine in the body is mast cells; however, several other types of cells can also synthesize histamine, such as neurons and keratinocytes [7]. However, mast cells were not found in either of the injection sites of the mock- and PRV TJ-infected mice. Additionally, the HDC mRNA level of the skin was also quantified. There was no statistical difference between the infected and mock groups (*p* ≥ 0.05) (Figure 4B).

Chlorpheniramine, which is an H1R antagonist, is widely used as a first-generation sedative antihistamine [30,31]. The mice received an intramuscular injection of either chlorpheniramine (10, 30, or 60 mg/kg) or DMEM at 44 and 52 hpi, and their behaviors were recorded 30 min later. As shown in Figure 4C, the frequency of the bite bouts and the durations of itch of the mice treated with chlorpheniramine were reduced in a dose-dependent manner.

### 3.5. The Severity of Itch Was Different between the Three PRV Strains and Was Consistent with the HDC Expression

A mouse model of the itch caused by PRV TJ was elaborated, and the increased level of HDC was responsible for the itch. To verify the relationship between HDC and the itch induced by PRV, the mRNAs of the HDC in the mice infected with 10^5^ TCID_50_ of PRV Bartha-K61, SC or TJ were quantified. First, the severity of the itch in the mice infected with different PRV strains was evaluated. At 56 hpi, the bite bouts of the mice infected with PRV TJ and SC were about 55 and 56 times per 30 min, respectively. Although the itch severities of the mice infected with PRV TJ and SC were similar in terms of the bite bouts, the bite durations of the mice infected with the two strains were different (*p* < 0.05). The itch duration of the mice infected with PRV TJ was about 91 s per 30 min, which was significantly different from that of the PRV SC (about 67 s per 30 min). Moreover, the mice infected with PRV Bartha-K61 were mildly pruritic, both in terms of the bite bouts and the durations per 30 min (Figure 5A). The mice infected with different PRV strains were photographed in the prone position. As shown in Figure 5B, the mice infected with PRV TJ developed substantial necrotic lesions at the injection sites at 56 hpi. To the contrary, the mice infected with PRV SC were also moribund, but they did not have injuries. Because of the reduced virulence of PRV Bartha-K61, the mice showed no abnormal changes. Overall, the itch severities in the mice infected with the same doses of different PRV strains were in the following order: PRV TJ > PRV SC > PRV Bartha-K61, which presented a positive correlation with the viral virulence. As shown in Figure 5C, the replications of different PRV strains in the DRG neurons were quantified. The genome copies of PRV TJ, PRV SC, and PRV Bartha-K61 in the DRG neurons at 56 hpi were about 10^4.0^, 10^3.6^, and 10^4.3^, respectively. Therefore, the itch severity was not associated with the replication of different PRV strains.

The HDC mRNA levels in the DRG neurons from the mice infected with different PRV strains were quantified (Figure 5D). The mRNA level of the HDC in the DRG neurons that were isolated from the mice infected with PRV TJ, SC, and Bartha-K61 increased 25-, 14-, and 5-fold, respectively, compared to the mock group, and these values were proportional to the itch severities of the mice.

## 4. Discussion

Since the first case of mad itch was described in 1813, the characteristic itch caused by PRV infection in nonnatural hosts has been frequently reported. Currently, few studies have focused on this aspect of PRV pathogenesis. Self-mutilating behavior, such as biting and scratching, can be stimulated by PRV infection, which is attributed to tissue lesions [32]. It is well established that itch can be induced by nerve injury caused by virus replication in the PNS [29]. However, the exact mechanism remains unknown, and the itch in the mice induced by PRV has not been quantified. Here, we quantified the itch severity of the PRV TJ-infected mice in terms of the bite bouts and durations. In addition, we demonstrated that histamine contributed to the itch caused by the PRV in the mice. 

When mice are infected with PRV, itch evokes innate scratching. It was previously shown that viral replication in the skin did not result in necrosis; however, the itch behaviors of the mice were not recorded and quantified [32]. The mice were videotaped after PRV TJ inoculation, they started itching at 32 hpi, and their legs suffered skin injuries at the same time point. 

We described the itch phenotype of the mice infected with PRV TJ, and we quantified the transcriptional kinetics of the different receptors and channels expressed in the DRG. The neurotransmitters, receptors, and signal pathways that are involved in acute itch transduction have been well revealed recently. Mrgprs are present in certain kinds of sensory neurons, and they are associated with the transmission of itch signals [33,34].

According to the results, MrgprD and MrgprX1, rather than MrgprA3, expressed differently, compared to the mock group, which may mean that the neural excitation was associated with an increase in the sodium current in the DRG neurons that express MrgprD and MrgprX1 [33,34]. TRPM8, which is a cold-sensitive ion channel, is involved in itch relief [35,36]. Increased TRPM8 expression may inhibit itch signaling in the primary sensory. The TRPV1-gene-expression level significantly increased, which may be due to its important role in histaminergic itch. In conclusion, the transcriptional levels of different molecules were different, which provides a framework for future work on the examination of the neuronal mechanisms that underlie PRV-induced itch and its sensitization.

Although the invasion of PRV into nonnatural hosts has not been well studied, it has been revealed that PRV can invade the PNS and the CNS via anterograde axonal transport. The PRV-induced itch may be triggered by the replication of the virus in the neurons [29]. DRG neurons were isolated and tested for the presence of PRV TJ. As expected, the IHC results demonstrated the presence of PRV TJ in the DRG neurons. Pruritogens, including TAC1, IL-2, IL-31, TPSAB1, TPSB2, and TPSG1, and the key enzymes that are attributed to itch, including HDC and TPH1, were detected by RNA-seq. The mRNA level of HDC was markedly increased, and the HDC protein was strongly positive in the DRG neurons (Figure 3B,C). It was well proven that the PRV TJ replication in the DRG neurons led to the increased expression of the HDC protein. 

More importantly, the IHC for histamine was strongly positive in the skin (Figure 4A). Histamine can be secreted by several kinds of cells, including neurons, mast cells, and keratinocytes [7]. The mast cells in the skin were detected by hematoxylin and eosin staining, as well as the mRNA levels of the HDC in the keratinocytes, and all of them presented no differences compared to the mock group (Figure 4A,B). All the results above prove that activated HDC results in increased histamine level in the skin.

Histamine, which is produced in either an autocrine or a paracrine manner, is an inflammatory mediator that is synthesized by HDC. In non-mast cells, HDC is rarely expressed without stimulation; however, its expression is markedly induced by various types of inflammatory stimulants [37,38]. In the mice infected with PRV, neuroinflammation was found in the DRG neurons. Therefore, we speculated that the neuroinflammation induced by PRV resulted in increased expression of HDC, which is regulated by mitogen-activated protein (MAP) kinases and c-Jun N-terminal kinase (JNK), which are important immune-regulatory molecules [39,40]. Upon infection with PRV, the MAPK and JNK expressions are upregulated [41], which leads to the change in normal cellular activities. Moreover, PRV possesses a linear double-stranded DNA with GC-rich sequences [42]. The transcription factor SP1, which binds to GC-rich regions, also regulates the expression of HDC [43]. Thus, it is possible that certain segments of the PRV genome bind to Specificity Protein 1, which results in the upregulation of HDC. In summary, the cellular and metabolic processes in the DRG neurons will be altered upon viral infection. During these processes, multiple viral factors, such as viral DNA, RNA, and proteins, may play important roles, which need to be explored in details.

Histamine exerts various immune-regulatory functions, and it plays an important role in neuroinflammation [44]. The activation of H1R expressed in the neuron by histamine has the ability to induce pruritus and atopic dermatitis [45]. This may explain the itch exhibited by the PRV-infected mice. The H4R-mediated activation of mast cells that is induced by histamine leads to the expression of various proinflammatory cytokines and chemokines, such as IL-6 [46], which have been proven to increase the protein levels in mice after infection with PRV [47,48]. Moreover, the activation of H4R involves several signaling cascades for the release of various allergic inflammatory mediators. Histamine can induce the phosphorylation of ERK in order to mediate the proliferation, differentiation, anti-apoptosis, regulation, and cytokine expression at the gene level. It can also activate NF-κB via the JAK-STAT signaling pathway [49,50], which could be well activated in the DRG neurons from the mice infected with PRV [41]. Taken together, the PRV-induced itch may be due to the histamine-induced inflammation, which presents a positive correlation with PRV virulence, but has no association with the replication of different strains in the DRG neurons (Figure 5B,C).

Many viral proteins are associated with PRV virulence, such as US9, gE, and gI. The loss of these genes usually results in virus attenuation. PRV Becker is a virulent strain; however, when deleting the genes that encode gI, US9, and gE, the virus induced decreased itch in mice. The PRV Bartha-K61, which carries deletions in the above three genes, induced the greatest attenuation of itch, compared with the three PRV isogenic strains [32]. This indicates that the itch severity is related to the virulence-associated proteins of the PRV. In addition, the genes encoding the gI and gE proteins of PRV TJ and SC show some sequence differences [6], which might lead to differences in the severity of the itch [6]. Apart from the three proteins mentioned above, other viral proteins, such as US6, UL27, UL44, etc., are different among the three PRV strains. Therefore, we speculate that these proteins may affect the severity of itch.

In conclusion, we found that the high level of histamine produced in PRV-infected DRG neurons resulted in itch, and that the severity of the itch induced by the different PRV strains was proportional to the HDC mRNA level in the DRG neurons of the PRV-infected mice.

## Figures and Tables

**Figure 1 viruses-14-01067-f001:**
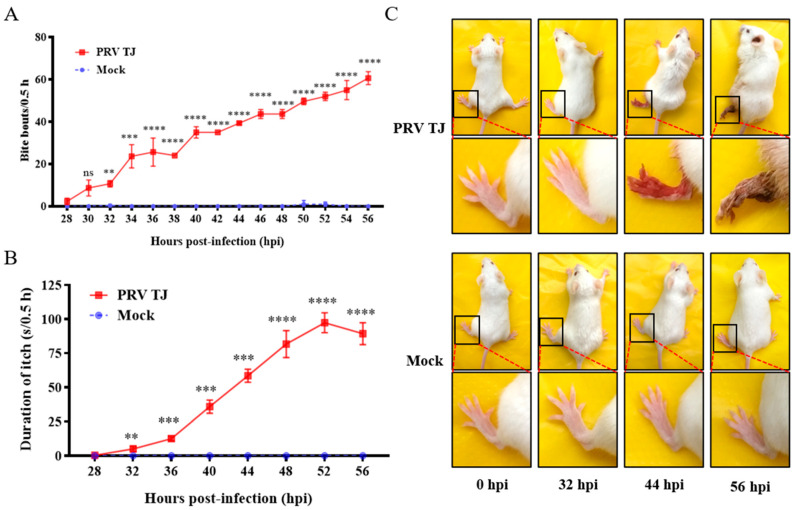
The mice infected with PRV TJ exhibited itch. Twelve SPF BALB/c mice were divided into two groups (*n* = 6): one group was infected intramuscularly with 10^5^ TCID_50_ of PRV TJ, and the other group was mock inoculated (100 μL of medium only). (**A**) The behaviors of the mice were recorded from 0 to 56 hpi. The bite bouts of the mice were recorded for 30 min at 1.5 h intervals. (**B**) The durations of biting were recorded for 30 min at 3.5 h intervals. (**C**) Representative photographs of the mice in the two groups were taken at 0, 32, 44, and 56 hpi. Bars represent the means ± SDs for three independent experiments; ns: not significant (*p*  ≥  0.05); ** *p* < 0.01; *** *p* < 0.001; **** *p* < 0.0001.

**Figure 2 viruses-14-01067-f002:**
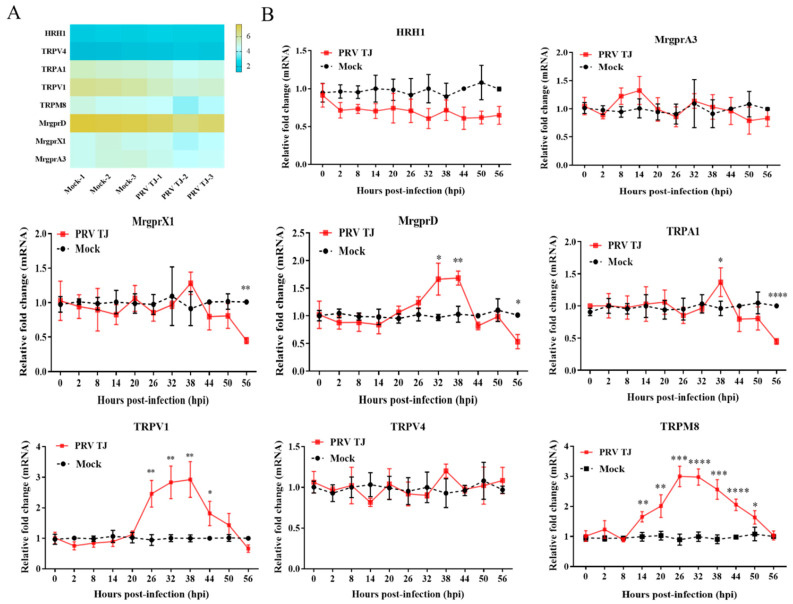
The mRNA expression levels of receptors and the potential cation channels relevant to itch-signal transmission were different. A total of 33 SPF BALB/c mice were infected intramuscularly with 10^5^ TCID_50_ of PRV TJ, and the same number of mice was injected with 100 μL of DMEM as a control. The DRG neurons on the left side of the spinal cord were collected, and total RNAs were extracted. (**A**) The mRNAs of HRH1, TRPV4, TRPA1, TRPV4, TRPM8, MrgprD, MrgprX1, and MrgprA3 at 56 hpi were quantified by using RNA-seq. (**B**) The expression kinetics were examined by using RT-qPCR. Bars represent the means ± SDs for three independent experiments; ns: not significant (*p*  ≥  0.05); * *p* < 0.05; ** *p* < 0.01; *** *p* < 0.001; **** *p* < 0.0001.

**Figure 3 viruses-14-01067-f003:**
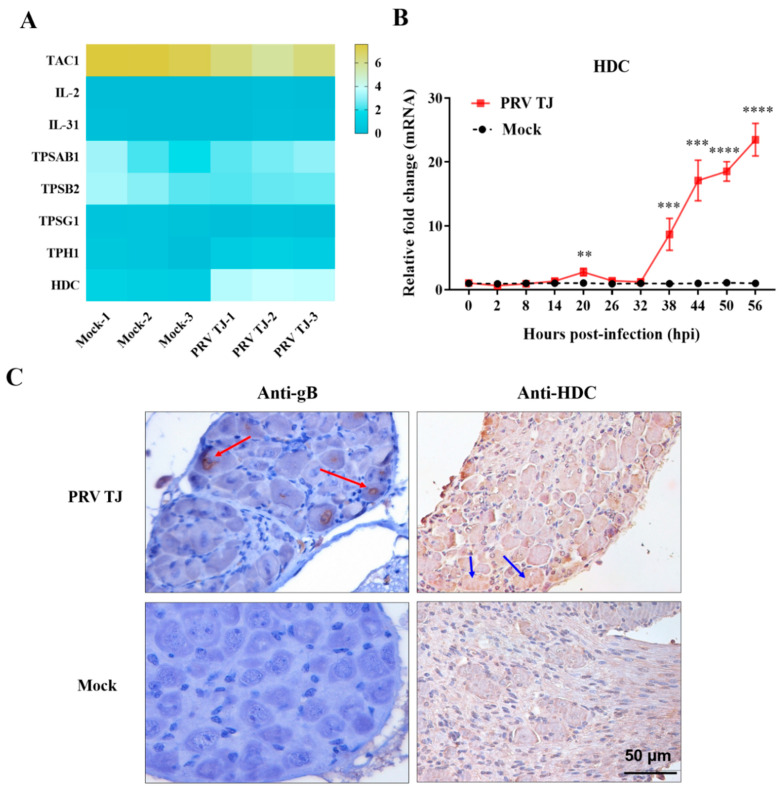
HDC was expressed in the DRG neurons of the PRV TJ-infected mice. Thirty-three SPF BALB/c mice were infected intramuscularly with 10^5^ TCID_50_ of PRV TJ, and the same number of mice was injected with 100 μL of DMEM as a control. The DRG neurons on the left side of the spinal cord were isolated, and total RNAs were extracted. (**A**) The mRNA expressions of TAC1, TPH1, IL-31, IL-2, HDC, TPSAB1, TPSB2, and TPSG1 were detected by using RNA-seq at 56 hpi. (**B**) The expression kinetics of HDC was detected by using RT-qPCR. (**C**) The cells expressing the gB protein of PRV (red arrows) and HDC in the lumbosacral DRG neurons (blue arrows) were detected by immunohistochemistry. Bar: 50 μm. Bars represent the means ± SDs for three independent experiments; ns: not significant (*p*  ≥  0.05); ** *p <* 0.01; *** *p* < 0.001; **** *p* < 0.0001.

**Figure 4 viruses-14-01067-f004:**
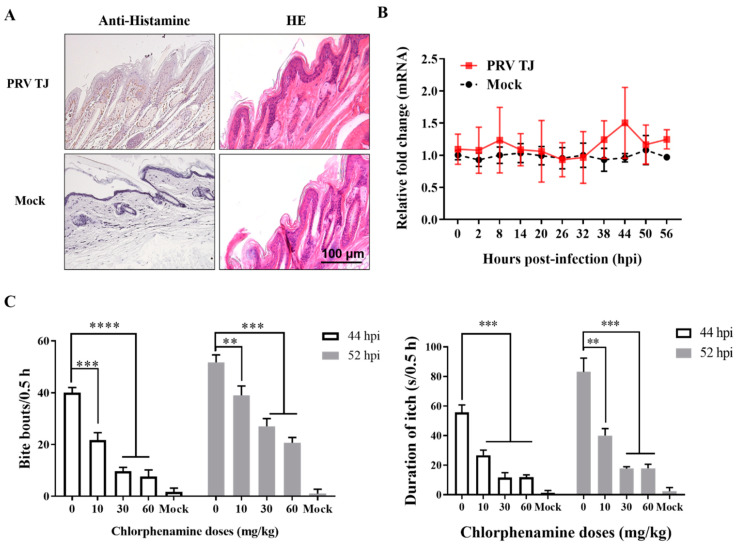
Histamine produced in the DRG neurons contributed to itch and could be inhibited by chlorpheniramine. Twelve SPF BALB/c mice were injected with 10^5^ TCID_50_ of PRV TJ or 100 μL of DMEM as a control. The PRV TJ-infected mice were treated with 0, 10, 30, or 60 mg/kg chlorpheniramine at 44 and 56 hpi, respectively, and the behavior of the mice was captured photographically after twenty minutes. The skin at the injection site was collected for pathological detection and total RNA extraction at 56 hpi. (**A**) The histamine was detected by the immunohistochemistry, and the mast cells were detected by using hematoxylin and eosin staining. Bar: 100 μm. (**B**) The mRNA expression trend of the HDC with time was quantified by using RT-qPCR. (**C**) The frequency of bite bouts and the durations of itch were counted after treatment with chlorpheniramine. Bars represent the means ± SDs of three independent experiments. Ns: not significant (*p*  ≥  0.05); ** *p* < 0.01; *** *p* < 0.001; **** *p* < 0.0001.

**Figure 5 viruses-14-01067-f005:**
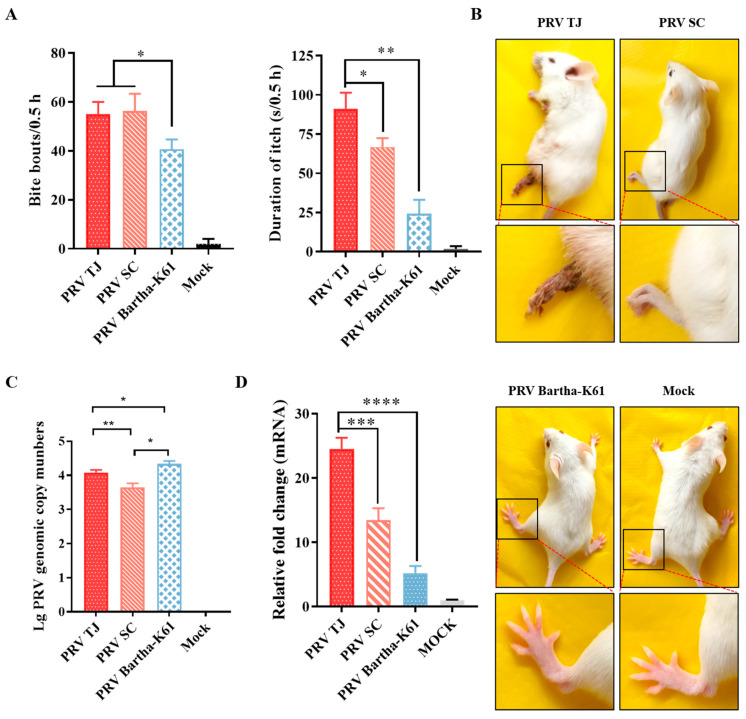
The severities of itch and HDC expressions of the three PRV-strain-infected mice. Twenty-four SPF BALB/c mice were divided into four groups. Six mice were infected with 10^5^ TCID_50_ of PRV TJ, PRV SC, PRV Bartha-K61, or DMEM only as a control. The behaviors of the mice infected with the abovementioned strains were recorded at 56 hpi. The DRG neurons of the mice in every group were isolated at 56 hpi, and the total RNAs were extracted. (**A**) The frequency of bite bouts and the durations of itch were recorded as above. (**B**) Representative photographs of the mice in the above mentioned groups taken at 56 hpi. (**C**) Genomic copies of different PRV strains in the DRG neurons of the infected mice at 56 hpi. (**D**) The DRG neurons of the mice in each group were isolated at 56 hpi, the total RNAs were extracted, and the expressions of HDC in the DRG neurons were quantified by using RT-qPCR. Bars represent the means ± SDs for three independent experiments; ns: not significant (*p*  ≥  0.05); * *p* < 0.05; ** *p* < 0.01; *** *p* < 0.001; **** *p* < 0.0001.

**Table 1 viruses-14-01067-t001:** Primers for RT-qPCR.

Primers	Sequences (5′-3′)
HRH1-F	ACTTGAACCGAGAGCGGAAG
HRH1-R	TTGCACAGCGGGTAGATGAG
TRPV4-F	TCACCCTCCTGAATCCGTGC
TRPV4-R	TCTCACCCATGAGGGCGAT
TRPA1-F	GGAAGTAATTCCTTTTCAGAGTGTC
TRPA1-R	ACTCCTCAACCACCCTGTGT
TRPV1-F	ACCACGGCTGCTTACTAT
TRPV1-R	AACTCTTGAGGGATGGTC
TRPM8-F	TACTCTGGCAGCCTTGGG
TRPM8-R	TCGCAGGAGTAGACCAGTAG
MrgprD-F	ATGAACTCCACTCTTGAC
MrgprD-R	AGCACATAGACACAGAAG

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
