# Peer review of "Histamine Is Responsible for the Neuropathic Itch Induced by the Pseudorabies Virus Variant in a Mouse Model"

_viruses, 2022, doi:10.3390/v14051067_

Round 1
Reviewer 1 Report
Pseudorabies virus (PRV) can cause severe acute neuropathy and the so-called “mad itch” in non-natural hosts. Little attention has been paid to the underlying mechanism of the itch caused by PRV in non-natural hosts. In this study, Wang et al. developed a mouse model of the itch caused by PRV. The itch-associated bite behavior, frequency, and duration were recorded and quantified. The authors confirmed that histamine is responsible for the neuropathic itch caused by pseudorabies virus. Generally, the work is something interesting. However, there are several concerns to be addressed.
- Why choose PRV TJ strain to construct the itch model? Please introduce the background information of PRV TJ strain.
- How to distinguish a natural bite from an itchy bite?
- Is this itch model reproducible?
- Whether the severity of itching reflects the virulence of PRV strains?
- At the protein level, whether histamine expression was different?
- More references were need to introduce histamine induced-itch.
- The description of figure legend in Fig.5 is not correct. The description of Fig. 5C should match the Fig.5B, and description of Fig. 5B should match Fig.5C.
- I have not found the detailed use of the Indirect Immunofluorescence Assay (IFA) method described in 2.2 throughout the paper. What is it used for in this paper?
- What is referred by the word of“Patents”in line 387?
Reviewer 2 Report
Pseudorabies virus (PRV) causes mad itch in non-natural hosts, but the mechanism is unclear. In this manuscript, the authors have addressed the issue using a mouse model. They found that transcription level of histidine decarboxylase (HDC), catalyzing the from histidine to histamine, was up-regulated in the dorsal root ganglion (DRG) of infected mice. The administration of chlorpheniramine, the H1 histamine receptor blocker could improve the itch behavior in infected mice.
It is not so surprising that histamine plays an important role in mad itch caused by PRV, but this manuscript may useful to understand the PRV pathogenesis.
I have some concerns in this manuscript as follows:
- Although they conducted RNA-seq at 56 hpi, I think they should have performed earlier phase. Because it is expected that more changes in gene expression would have occurred and some of them induced HDC mRNA expression. The mechanism of the HDC gene expression induced by PRV infection is still unclear. The authors should discuss this point.
- The authors also have compared the severity among three strains. They should show whether the severity is correlated to the viral replication level in DRG.
- Photos of IHC analyses in Figs 3C and 4A are unclear.
A minor point:
Fig legends for Fig 5B and 5C are interchanged.
Round 2
Reviewer 2 Report
1. Although they conducted RNA-seq at 56 hpi, I think they should have performed earlier phase. Because it is expected that more changes in gene expression would have occurred and some of them induced HDC mRNA expression. The mechanism of the HDC gene expression induced by PRV infection is still unclear. The authors should discuss this point.
Author’s response: At 56 hpi, the mice infected with PRV showed the most severe clinical signs, therefore RNA-seq was conducted. The regulation of histamine in vivo has not been clarified so far. Histamine, produced in either autocrine or paracrine manner, exerts various immune regulatory functions and plays important roles in neuroinflammation. We have discussed this point (Lines 409-428).
Unfortunately, the authors did not adequately answer to my comment. The upregulation of ‘HDC’ mRNA level was detected at 38 hpi (Figure 3B), so host genes related to the induction of HDC mRNA expression may have been identified if they have conducted RNA-seq at around 38 hpi. I think this point is a weakness of this work. The regulation of HDC mRNA expression is somewhat known (https://www.ncbi.nlm.nih.gov/pmc/articles/PMC6359378/). Moreover, additional data indicates that the viral replication level in the DRG was not different between infections of three PRV stains although the severity was different. If so, the authors should discuss what viral factors are responsible for the differences in HDC mRNA expression.
3. Photos of IHC analyses in Figs 3C and 4A are unclear.
Author’s response: Figures 3C and 4A have been updated.
These figures have been updated but these are still difficult to understand. For Figure 3C, which DRGs were used for this analysis? Lumbosacral region? Please state. It appears that different sites were used. In addition, the authors should explain what the arrows indicate. For Figure 4A, captions for images are disappeared. The authors described “But mast cells were not found in both injection sites of the mock- and PRV TJ-infected mice“ in the main text, but “the mast cell were observed using hematoxylin and eosin staining” in the legend. In addition, although lower panels are considered to be derived from the mock infection, skin structures appear to be different. Why?
